# Implementation of Elements of the Concept of Lean Construction in the Fire Protection of Steel Structures at Oil and Gas Facilities

Marina Gravit , Nail Ikhiyanov, Anton Radaev and Daria Shabunina *

Peter the Great St. Petersburg Polytechnic University, 195251 Saint-Petersburg, Russia
* Correspondence: shabunina.de@edu.spbstu.ru

**Abstract:** The work is aimed at developing a procedure for applying the concept of lean construction to improve the technological process of applying fire protection coating on the steel structures of oil and gas facilities. The experience of implementing elements of the concept of lean construction in the activities of the organization of the oil and gas complex is presented. The developed procedure involves the use of elements of the concept of lean construction, such as value stream mapping, the "Spaghetti" diagram, and timekeeping elements of the technological process. For the example of an existing object of the oil and gas facility, the developed methodology for the implementation of the concept of lean construction is realized. The results of implementation showed that the output per worker increased by 33%, the process time for applying fire protection epoxy coating decreased by 35%, and the total distance of the route of workers in the process of applying the composition decreased by 19%. The practical significance of the results in this study consists of the possibility of using the developed procedure in the activities of construction organizations to improve technological processes.

**Keywords:** oil and gas facility; building structures; fire resistance limits; fire protection coating (FPC); epoxy coating; lean construction; procedure

## 1. Introduction

Fire safety is one of the most important operational characteristics of construction projects, including oil and gas facilities. Ensuring the fire safety of oil and gas facilities involves the development of various structural and technological solutions in the field of protection against threats of technogenic and environmental nature. The stability of structural systems of objects in case of fire and ensuring their required fire resistance is the most important factor for ensuring fire safety [1–3]. This circumstance is due to a significant deterioration in the strength properties of steel structures used in the oil and gas facilities in conditions of elevated temperatures. It determines the importance of using various fire protection coatings as part of coating systems in conjunction with the primer and finish coatings [4].

The high importance of the use of fire protection coating (FPC) for various steel elements of the production facility at the oil and gas complex is determined by the following features:

- The presence at the facility of large volumes of highly flammable and explosive products, in the case of ignition of which it is possible to develop a hydrocarbon fire regime, accompanied by a sharp jump in temperature and excessive pressure [5–8];
- High risk of loss of stability of loaded steel structures located in the place of localization of the fire due to a significant reduction in the strength of these structures at a temperature range of 400–600 °C (in the absence of fire protection);

    — The presence of high requirements imposed on elements of passive fire protection in relation to building structures as part of the oil and gas facilities [9,10], including the ensuring of the fire resistance limit of loss of strength as R120 [11].

These factors determine the need for a rational organization of the technological process of applying FPC onto metal structures in the oil and gas facilities. This technological process has the following main features [12,13]:

    — High labor intensity: the corresponding share (relative to the total value for all types of work) value of the labor cost indicator is about 30%;

    — Use of expendable material: fire protection composition and reinforcing glass mesh, with the volume determined by the operating characteristics of the treated metal structures;

    — Participation of a certain number of workers to ensure the parallel execution of individual elements of the process;

    — Use of technological equipment and tools for cleaning surfaces (sandblaster), caulking hard-to-reach places (hacksaw), and application of fireproofing composition on the surface (airless spraying unit) of metal structures.

The general view of steel structures with an FPC, used as part of the oil and gas facilities, is shown in Figure 1.

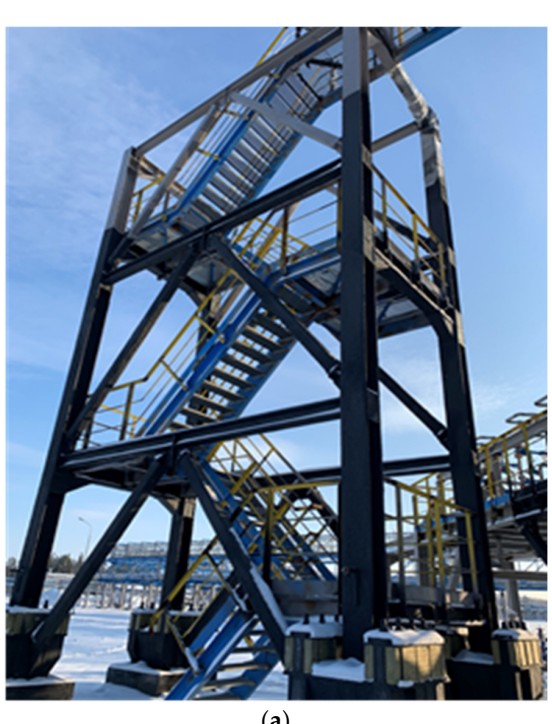 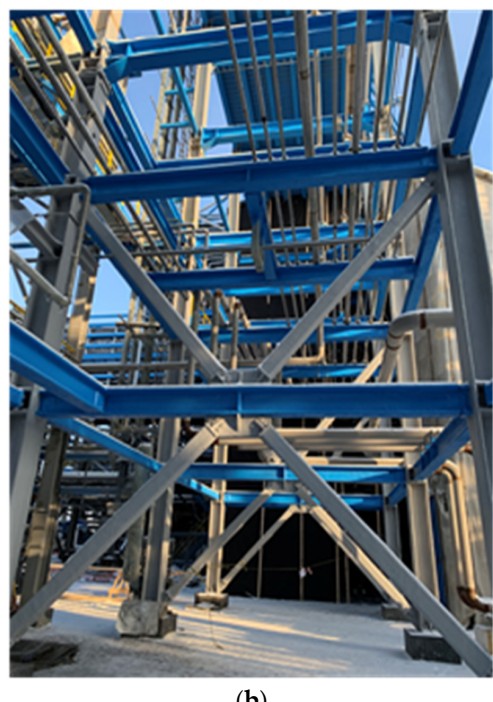

(**a**)                            (**b**)

**Figure 1.** General view of steel structures with an FPC used for oil and gas facilities: (**a**) general view of the rack and (**b**) element of the overpass with epoxy-based FPC.

The complexity of the structure of the technological process of applying FPC on metal structures determines the existence of a limited set of methodological developments and tools. The development of techniques and procedures will provide an effective (in terms of labor costs and the rationality of the obtained results) justification for process characteristics in its design or improvement.

The main problems of both technological processes and the oil and gas industry as a whole are the long-term implementation and inefficient use of resources, which leads to a decrease in the efficiency of work. These problems can be solved with the help of methodological developments and tools to accelerate the technological process in the stage of construction and operation of the object [14].

The concept of lean construction, which emerged in the early 1990s, occupies a special place among the already mentioned methodological developments and tools. Lean construction includes various elements of the concept of lean production developed in the middle of the previous century, adapted to the conditions of the implementation of various technological processes in the construction industry [14–17]. In particular, the concept of lean construction involves the use of the following categories of elements:

— Methodological developments that are applied in relation to the organizational structure of the enterprise and are intended to improve the efficiency of labor resources involved in the implementation of technological processes of construction (e.g., "Kaizen", "TQM", etc.) [18,19];

— Practical developments that are applied directly to the technological processes and are intended to reduce or eliminate various categories of losses in the implementation of certain elements of technological processes of construction (e.g., "Kanban", "5S", "SMED", etc.) [20–22];

— Tools of qualitative and quantitative description that are applied directly to technological processes and are intended to obtain information about the current and predictive (taking into account the application of methodological and practical developments) structure of technological processes.

Currently, the concept of lean construction is being implemented in construction projects to improve the qualitative and quantitative performances of the project. A study [23] gives an insight into the implementation of lean construction in Morocco. Lean construction is defined as providing innovative ways to manage construction projects, reducing losses and increasing efficiency. Another study [24] aims to assess the awareness and potential benefits of lean construction in Ethiopia. It is demonstrated that increased productivity, reduced losses, increased customer satisfaction, reduced project time, and high-quality construction are the benefits of implementing the concept of lean construction. In [25], a conceptual model was developed and verified using a systematic dynamics approach to consider the impact of lean construction on construction safety. The study in [26] proposes a methodology for improving assembly for an oil and gas facility by integrating building information modeling (BIM) and lean construction. As a result, the proposed methodology reduced the welding time by 8.7% compared to the global average assembly rate in construction projects. Nevertheless, the construction industry faces significant challenges in its development towards sustainability and the concept of lean construction on construction projects. In [27], the current state of lean construction implementation in the construction industry of the Kingdom of Saudi Arabia was studied. It was found that lack of management support, low awareness of lean construction, and lack of training are concomitant factors that prevent the implementation of the concept of lean construction. In [28], the current level of awareness of lean construction was assessed and the potential benefits and challenges of implementing lean construction in the construction industry of Bangladesh were identified. It was found that the lean construction approach has a positive impact on quality, safety, cost, productivity, and environment. The most significant problems in the implementation of lean construction are lack of awareness and skills, poor management, traditional culture and attitudes of workers, lack of resources and equipment, as well as the non-use of modern methods and technologies.

Thus, a sufficiently large number of studies on the successful implementation of the concept of lean construction in the activities of construction organizations as well as the enterprises of the oil and gas facilities (including "Exxon Mobil" (USA) [29], "Royal Dutch Shell" (Netherlands-British) [30], Saudi Aramco (Saudi Arabia) [31], "Tatneft" (Russia) [32], Norsok (Norway) [33], and TotalEnergies (France) [34]). Nevertheless, the analysis of scientific papers testified to the lack of developments that contain a description of the process of applying elements of the concept of lean construction to improve the technological process of applying FPC on the surface of metal structures on the objects of the oil and gas facilities.

The purpose of the study is to develop methodological developments and tools in the application of elements of the concept of lean construction to improve the technological process of applying FPC on the surface of metal structures for the oil and gas facility.

In accordance with this purpose of the study, the following tasks are formulated:

1. Review and analysis of scientific papers on the implementation of the concept of lean construction in the activities of industrial enterprises, including oil and gas facilities;

2. Development of a procedure for applying elements of concept of lean construction to improve the technological process of applying the FPC on the surface of metal structures for the oil and gas facility;

3. Implementation of the developed procedure on a practical example.

The object of the study is the technological process of applying FPC on the surface of metal structures for the oil and gas facility, improved through the use of elements of the concept of lean construction. The technological process generally involves layer-by-layer and uniform application of FPC (Figure 2) across the surface of the protected structure with an intermediate drying procedure (duration of at least 4 h at ambient temperature +20 °C) conducted before applying each additional (starting from the second) layer. If the thickness of the FPC is 6 mm or more, a reinforcing glass mesh shall be placed over the uncured layer of FPC on the treated surface of all external corners (ribs) of the structure.

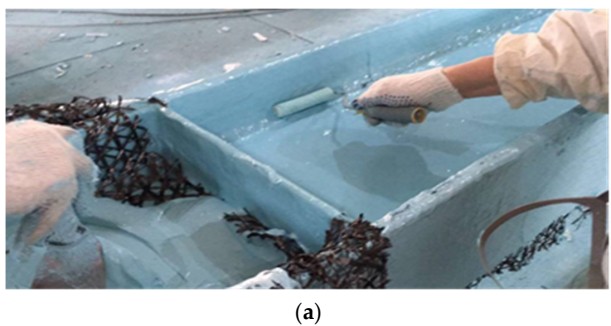　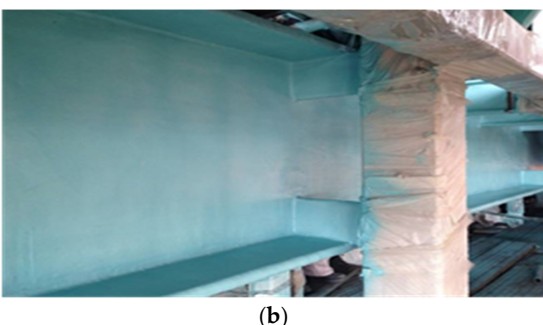

(**a**)　　　　　　　　　　　　　　　　　　　　　　　　(**b**)

**Figure 2.** (**a**) Process and result (**b**) of uniform and layer-by-layer application of the FPC on the surface of metal structures.

The subject of the study is the key (from the standpoint of the concept of lean construction) characteristics of this process as an object of study.

A detailed description of the developed procedure, as well as the results of its implementation on a practical example, are presented in the following sections of the work.

## 2. Materials and Methods

As part of the study, a procedure for applying elements of the concept of lean construction to improve the technological process of applying the FPC on the surface of metal structures for the oil and gas facility was developed. The main points of the developed procedure are presented below.

1. Application of elements of the concept of lean construction in relation to the considered technological process involves the formation of a certain set of actions, the combined implementation of which within a limited time interval provides an increase in the efficiency of implementation of technological operations. This is performed during the transition from the initial technological process (current state) to the changed process (future state) by reducing or eliminating the following categories of losses: overproduction, waiting, stocks, unnecessary movements of workers, unnecessary (from the position of satisfaction of customer requirements) process elements, and unused potential of labor resources.

2. To analyze the initial technological process, as well as to predict the changed technological process, it is advisable to use the following elements of qualitative and quantitative description provided within the concept of lean construction:

— Value stream mapping, which involves the division of the main elements of the technological process into the following categories: elements that add value to the process

result and elements that do not add value to the process result, further subdivided into non-excludable and potentially-excludable;

— "Spaghetti" diagram, which is a spatial network that includes the following elements: nodes that define locations, each of which performs certain elements of the technological process, and arcs (connections) between nodes that describe the trajectory of workers between locations;

— Timekeeping, which involves direct measurement of the duration of workers' performance on individual elements of the technological process.

These tools provide identification of deficiencies in the initial technological process as well as a qualitative assessment of the effect of the set of actions manifested within the changed technological process.

3. The key characteristics of the technological process necessary to objectively assess the effectiveness of the elements of the concept of lean construction are the following:

— Total length of workers' movements during the implementation of the technological process;

— The total duration of the elements of the technological process in the context of the categories provided for in the value stream mapping procedure;

— The total duration of the implementation of the technological process.

4. The measures necessary to improve the technological process are formed on the basis of the results of the analysis of the initial technological process considering the methodological and practical developments available for use as elements of the concept of lean construction.

5. The main characteristics of each individual activity that determine its effectiveness are the following:

— The duration of the action;

— Capital (one-time) and periodic costs necessary to implement the action;

— Reduced costs required to implement the event, which are calculated on the basis of capital and periodic costs, considering the forecast period of availability of the effect from the implementation of the action;

— The change in the length of labor movements in the transition from the initial technological process to the changed in accordance with the action;

— The change in the total duration of the elements of the technological process in the context of the categories provided for in the value stream mapping procedure;

— The change in the total duration of the implementation of the technological process.

6. The main characteristics of the set of actions, determining its efficiency, are calculated on the basis of the corresponding characteristics for individual actions using certain functional dependences, assuming, among other things, the additivity of the characteristics of the set of actions (for example, for periodic and capital costs).

The structure of the developed procedure is defined by a flowchart, as shown in Figure 3.

In the initial (first) stage of the procedure, information is collected and processed about the initial technological process in terms of its structure, the involved departments of the enterprise, the workers involved, and the technological equipment and tools used.

In the second stage of the procedure, the initial technological process is analyzed by applying the elements of qualitative and quantitative descriptions in relation to the information prepared at the previous stage, provided within the concept of lean construction, and followed by the calculation of the key characteristics of the process. Based on the obtained information, the identification of process deficiencies associated with the presence of different categories of losses allocated within the concept of lean construction.

The third stage of the procedure is the formation of alternative actions to improve the technological process. Formation is based on the results obtained during the previous stage of the procedure with the application of methodological and practical developments provided by the concept of lean construction. For each formed alternative action, the calculation of its main characteristics is made.

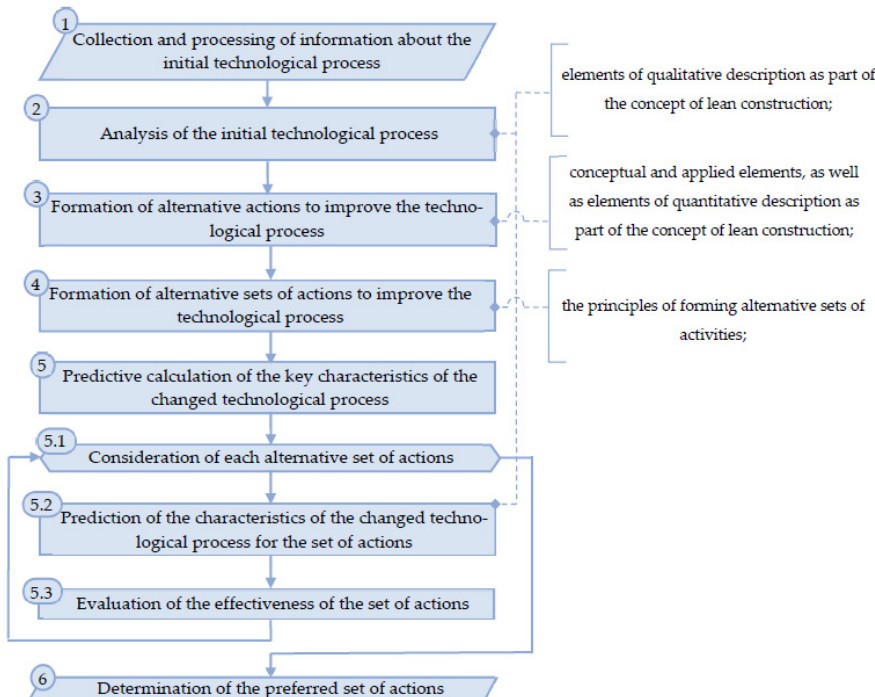

**Figure 3.** Flowchart describing the structure of the developed procedure.

In the fourth stage of the procedure, alternative sets of actions are generated based on the following constraints:

— The calculated value of the total duration of the actions in the set should not exceed the maximum permissible value;
— The calculated value of the total capital and (or) periodic costs for the implementation of actions in the set should not exceed the maximum permissible value.

The maximum permissible values, as part of the abovementioned constraints, are determined by the amount of production resources available to the organization to perform work on improving the technological process.

In the fifth step of the procedure, a predictive calculation of the key characteristics of the changed technological process is made, and on their basis, the calculation of the efficiency characteristics for each individual set of actions is performed.

In the final stage of the procedure, based on the calculation results obtained at the previous stage of the procedure, the preferred set of activities is determined. The preferred set of actions corresponds to the maximum value of the integral efficiency indicator, as which it is proposed to consider the ratio of the change in the total duration of the technological process to the total present value of the cost of implementing a set of actions.

Hence, the developed procedure is a sufficiently detailed, formalized description of the process of applying the elements of the concept of lean construction to improve the technological process of applying an FPC on the surface of metal structures for the oil and gas facility.

## 3. Results and Discussion

To assess the practical significance of the developed procedure during the final stages of the study, its implementation was performed on a practical example of an object of the oil and gas facility. The concept of lean construction is implemented in the activities of the technological engineering holding. It performs construction and installation works during the creation of a complex for production, storage, and shipment of liquefied natural gas as part of the compression station "Portovaya" (Vyborg, Leningrad Region, Russia). The main purpose is to improve the technological process of applying FPC on the metal structures of racks and overpasses from the base to the top to ensure the required fire resistance limit of

R120 according to the requirements of SP 4.13130.2013 [11] and special technical conditions for design and construction in terms of ensuring fire safety of the considered object. The general view of the infrastructure facilities of the compression station "Portovaya" is shown in Figure 4.

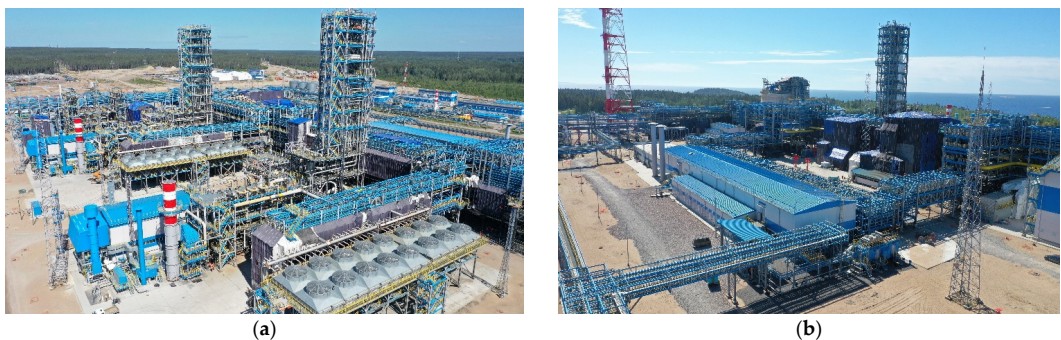

**Figure 4.** Infrastructure facilities at the compression station "Portovaya": (**a**) natural gas liquefaction department and (**b**) equipment building with a complete transformer substation.

Two-component epoxy coating (first component—disperse system based on epoxy binder, second—hardener) was used as an FPC, applied layer by layer to the primed surface. The scheme and general view of metal structures with FPC are shown in Figure 5.

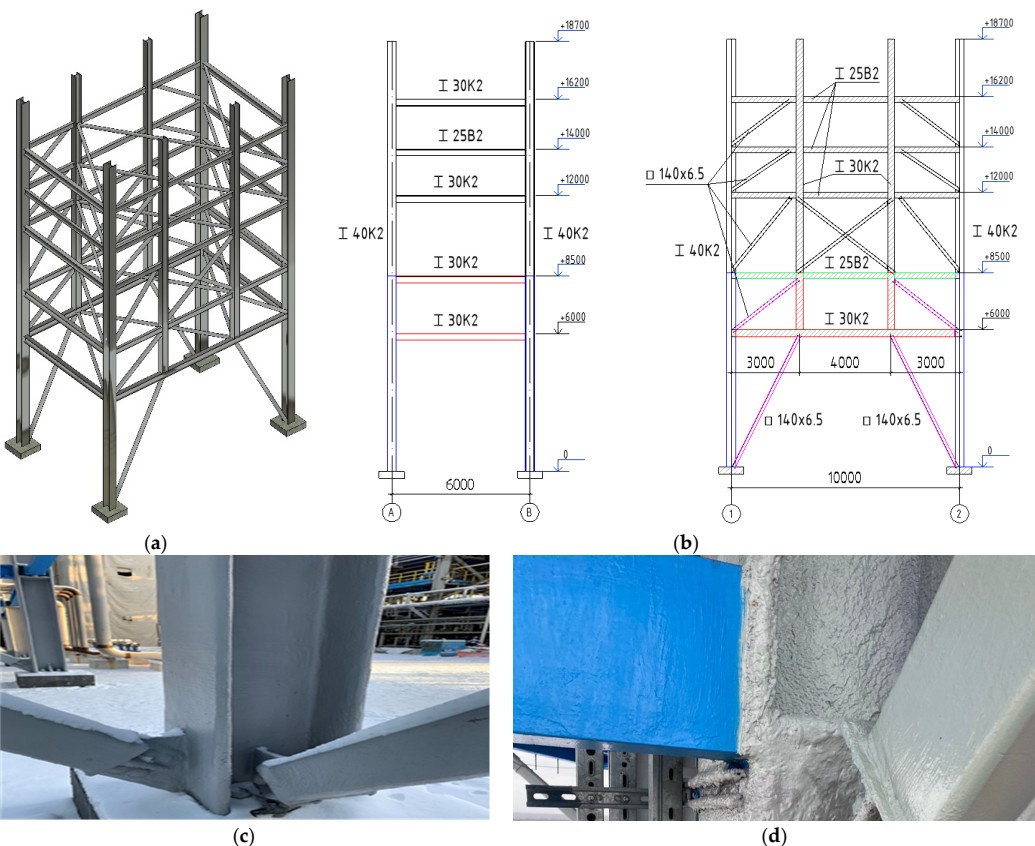

**Figure 5.** Steel structures with FPC: (**a**) model of a rack; (**b**) scheme of application of FPC on the surface of steel structures (the color determines the thickness of the FPC); (**c**) general view of the steel structure assembly; (**d**) fragment of the structure containing the junction of the rod elements.

The work was performed within 299 days from 4 January 2021 to 30 October 2021, according to a previously developed detailed schedule.

Information about the initial technological process as well as the results of the application of quantitative and qualitative descriptions (provided within the concept of lean construction) in relation to the initial technological process (the first and second stages of the procedure) are presented in Tables 1–3.

**Table 1.** The works on providing corrosion protection and application of FPC on the surface of steel structures.

| No. | Operation | Scope of Work, m$^2$ | Share in the Total Amount of Work,% | Remaining Scope of Work on 1 January 2021, m$^2$ |
|---|---|---|---|---|
| 1 | Restoration of corrosion protection | 12,846.42 | 56 | 5648.32 |
| 2 | Application of FPC | 100,194.96 | 23 | 77,298.71 |
| 3 | Application of the finishing layer of FPC | 221,051.42 | 1 | 219,718.12 |
| 4 | Restoration of the FPC | 3278.95 | 16 | 2743.47 |
| 5 | Corrosion protection of steel structures | 4290.07 | 18 | 3518.02 |

**Table 2.** The procedure of work for application of FPC in accordance with the structure of the initial technological process with the participation of the quality control service (QCS).

| No. | Procedure of the Technological Process | Number of Workers | Technological Equipment and Tools | Duration of Operations (the Result of Timekeeping) on 18 m$^2$, min | | |
|---|---|---|---|---|---|---|
| | | | | Operations That Add Value | Operations That Determine Losses | All Operations |
| 1 | Abrasive cleaning | 2 | Sandblaster | 276 | 509 | 785 |
| 2 | Handing over the cleaning to the QCS | 2 | | 20 | 160 | 180 |
| 3 | Priming | 1 | Airless spraying unit "Wagner 950E" | 80 | 370 | 450 |
| 4 | Handing over the primer to the QCS | 1 | | 25 | 125 | 150 |
| 5 | Caulking of hard-to-reach places | 1 | Measuring tape, hacksaw | 67 | 355 | 422 |
| 6 | Application of 3–5 mm layer of FPC | 1 | Airless spraying unit "Wagner 950E" | 449 | 960 | 1409 |
| 7 | Handing over 3–5 mm layer of FPC | 2 | | 60 | 1140 | 1200 |
| 8 | Installation of reinforcing mesh | 3 | Measuring tape | 66 | 54 | 120 |
| 9 | Handing over the reinforcing mesh to the QCS | 1 | | 30 | 30 | 60 |
| 10 | Application of an 8 mm layer of FPC | 2 | Airless spraying unit "Wagner 950E" | 1712 | 427 | 2139 |
| 11 | Handing over the FPC to the QCS | 1 | | 20 | 10 | 30 |
| 12 | Application of the finish layer | 2 | Airless spraying unit "Wagner 950E" | 220 | 20 | 240 |
| 13 | Handing over the finish layer | 1 | | 20 | 60 | 80 |
| | Summary: | 17 | | **3045** | **4220** | **7265** |

Figure 6 shows the results of value stream mapping for the initial technological process. The flow mapping allows to trace the production chain of the FPC process, starting with the acceptance of metal structures and delivery of the finished result to the customer. The value stream mapping takes into account the main operations associated with the fire protection application process.

**Table 3.** The results of timekeeping the elements of the initial technological process.

| Categories of Process Operations | The Duration of Operations within the Procedure of the Technological Process with an Ordinal Number (See Table 2) | | | | | | | | | | | | | TOTAL Duration of Operations in the Process |
|---|---|---|---|---|---|---|---|---|---|---|---|---|---|---|
| | 1 | 2 | 3 | 4 | 5 | 6 | 7 | 8 | 9 | 10 | 11 | 12 | 13 | |
| All operations | 720 | 180 | 450 | 150 | 422 | 1409 | 1200 | 120 | 60 | 2139 | 30 | 240 | 80 | **7265** |
| Operations that add value | 276 | 20 | 80 | 25 | 67 | 449 | 60 | 66 | 30 | 1712 | 20 | 220 | 20 | **3045** |
| Operations that do not add value | 509 | 160 | 370 | 125 | 355 | 960 | 1140 | 54 | 30 | 427 | 10 | 20 | 60 | **4220** |

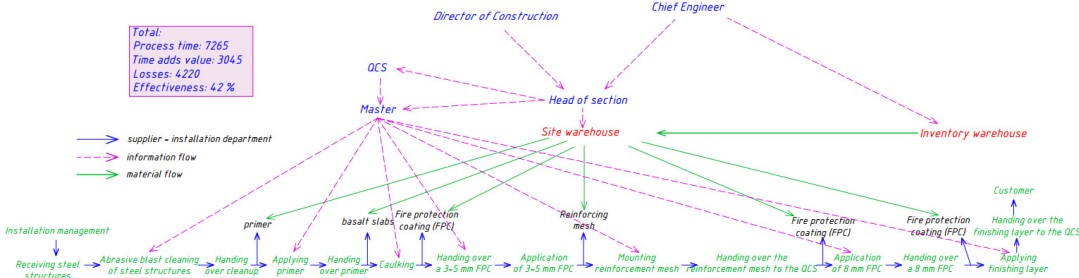

**Figure 6.** Results of value stream mapping for the initial technological process.

Figure 7 shows the results of the "Spaghetti" diagram for the individual elements of the initial technological process. The "Spaghetti" diagram allows us to visualize the physical movement and distances involved in the process. By analyzing the movement map, the potential for speeding up and simplifying all operations associated with the process of applying fireproofing is identified. The "Spaghetti" diagrams are made for each operation, starting with the acceptance of metal structures and the delivery of the finished result to the customer.

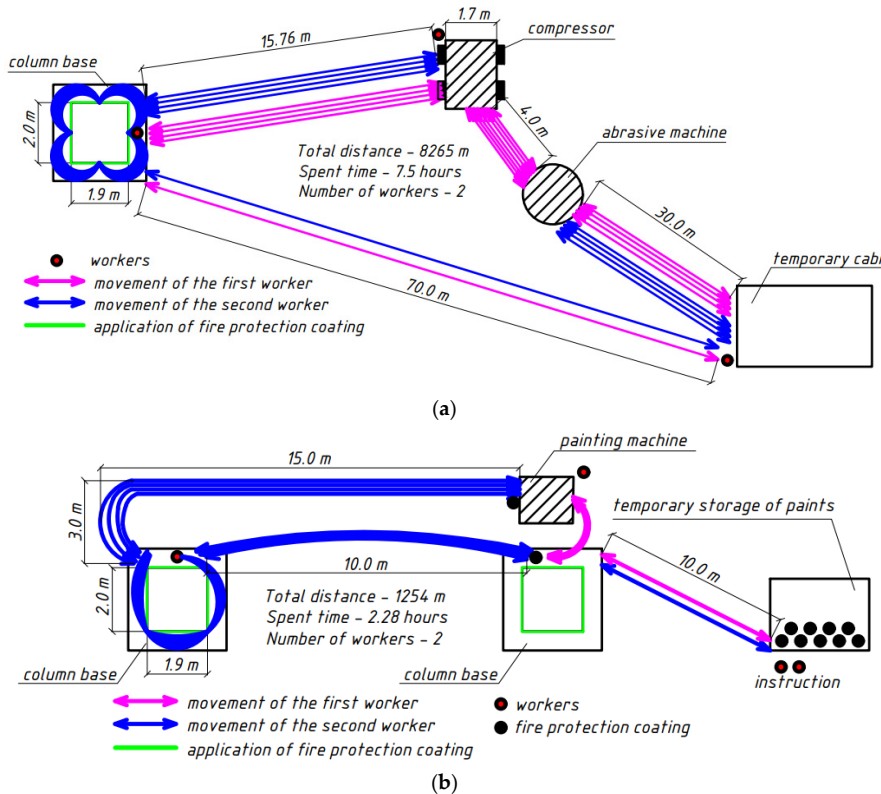

**Figure 7.** *Cont.*

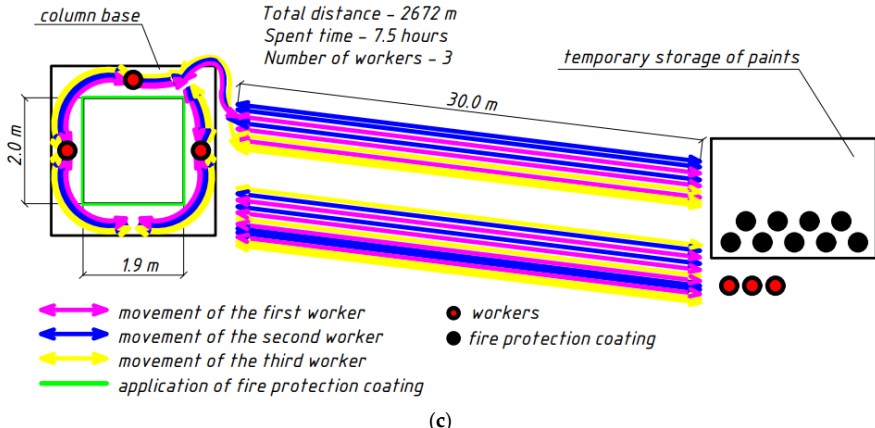

(**c**)

**Figure 7.** Results of drawing the "Spaghetti" diagrams for individual elements of the initial technological process: (**a**) procedure of abrasive cleaning; (**b**) procedure of single-layer application of fire protective epoxy coating; (**c**) procedure of supplying places of work with fire protective epoxy coating from the area of temporary storage of coating materials.

The following deficiencies of the initial technological process are identified:

— The presence of a sufficiently high rate of defective products detected within the QCS (18%), which determines the additional losses for the elimination of defects;
— Inefficient organization of procedures for preparing workstations and technological equipment, its washing and cleaning, as well as procedures for cleaning workstations and delivery of consumables, which causes additional time losses;
— The presence of repetitive movements of workers between locations performing the operations of the technological process, which determines a sufficiently large value of the total length of movement—10,524 m.

The importance of eliminating these deficiencies is confirmed by the results of time-keeping of the initial technological process (Table 3):

— The total duration of the process is 7265 min, including 3045 min as the duration of operations that add value and 4220 min as the duration of operations that do not add value;
— The distribution of the duration of operations that do not add value by loss categories is as follows: moving—59%, waiting—10%, transportation—8%, rework—20%, and excessive processing—3%.

The description of the formed set in the third stage of the procedure alternative measures, the implementation of which will reduce or eliminate the negative impact of the above deficiencies on the effectiveness of the implementation of the technological process, is presented in Table 4. Table 4 also contains information about the compliance of these measures with the alternative set (a total of three) formed in the fourth stage of the procedure in accordance with the following constraints:

— The maximum allowable duration of the activities in the set is 1250 h;
— The maximum allowable value of the total capital costs for the implementation of activities in the set is 1800 monetary units;
— The maximum allowable value of the total current (periodic) costs for the implementation of the activities in the set is 130 monetary units/period. In the process of implementation of the procedure, EUR 16 was considered as the equivalent of 1 monetary unit, and a month was considered as the calculation period.

The results of the predictive calculations of the characteristics of the changed technological process for each of the sets of actions, performed in the fifth stage of the procedure, are presented in Table 5.

**Table 4.** Description of alternative activities and the sets formed on their basis.

| No. | Location of the Problem | Description of the Problem | Name of Action | The Sequence Number of the Set of Actions |
|---|---|---|---|---|
| 1 | Construction site | Expectation of QCS | Making an acceptance schedule daily | 2 |
| 2 | Fire protection area | Insufficient qualification of employees | Drawing up a competency matrix, implementation of mentoring | 1, 2, 3 |
| 3 | Construction site | Lack of control during night shifts | Delegation of authority to supervisors | 1, 2, 3 |
| 4 | Place of FPC application | Sudden stoppage of processes due to tests, reassignment of workers to other areas of work | Planning work in accordance with the test schedule to avoid organizational and technological interruptions and the movement of workers to other areas | 1 |
| 5 | Place of FPC application | Inefficient tool storage | Efficient tool storage | 2 |
| 6 | Warehouse | Long receiving overalls | Development as part of a complementary project | 2 |
| 7 | Supply management | Untimely ordering of materials from suppliers | Development as part of a complementary project | 3 |
| 8 | Place of caulking and installation of mesh | Inefficient measuring tools | Using a laser tape measure | 1, 2, 3 |
| 9 | All sites | Repeated cutting with a hacksaw | Changing the hacksaw to a carbon steel knife | 1, 2, 3 |
| 10 | All sites | Ineffective work of masters on quality control | Introduction of a manufacturing analysis of defects | 1, 2, 3 |
| 11 | Construction site | Adverse weather conditions | Covering work sites | 1, 2, 3 |
| 12 | Priming site | Redistribution of workers to fix defects | Timely detection and correction of defects | 1 |
| 13 | All sites | High time costs due to the variability of technological operations | Development of standard operating cards | 1, 2, 3 |
| 14 | Temporary storage warehouse | Storage of materials in intermediate storage in large quantities | Calculation of the frequency of deliveries, reducing the volume of the order | 1, 2, 3 |
| 15 | Primer application site | Defects due to discrepancies in layer thickness | Implementation of a production analysis of defects | 1, 2, 3 |
| 16 | Sandblasting area | Inconsistency of the second degree of cleaning, the appearance of corrosion | Implementation of a production analysis of defects | 3 |
| 17 | Reinforcement mesh installation area | Defects due to inconsistency with the manufacturer's instructions | Implementation of a production analysis of defects | 1, 2, 3 |
| 18 | Finishing layer application area | Excessive time required to form a cant | Exclusion of operation | 1, 2, 3 |
| 19 | Sandblasting area | Stopping the process due to refilling of the abrasive blast machine | Using the second machine for abrasive blast cleaning; loading the first machine while the second one is working | 1 |
| 20 | FPC application area | Smoothing the surface for an aesthetic look | Exclusion of operation | 3 |

The first set of actions corresponds to the maximum value of the integral efficiency indicator, which is equal to the ratio of the change in the total duration of the implementation of the technological process to the total present value of the cost of implementing a set of actions. Thus, the first set of actions is identified as the most preferable in the final stage of the procedure. The activities in the specified set are implemented in relation to the initial technological process within the planned time limits.

**Table 5.** Prediction of the characteristics of the changed technological process for each set of actions.

| Workers' Index | Workers | Total Distance of Movements, m | Total Duration of Operations | | | |
|---|---|---|---|---|---|---|
| | | | Operations That Add Value, sec. | Non-Excludable Operations That Do Not Add Value, sec. | Potentially Excludable Operations That Do Not Add Value, sec. | All Operations, sec. |
| Initial technological process | | | | | | |
| 1 | First worker | 1027.64 | 12.91 | 12.32 | 10.76 | 35.98 |
| 2 | Second worker | 887 | 12.90 | 12.29 | 10.76 | 35.95 |
| Total value | | 1914.64 | 25.81 | 24.61 | 21.52 | 71.93 |
| Changed technological process (according to the first set of actions) | | | | | | |
| 1 | First worker | 986.36 | 12.16 | 6.44 | 8.92 | 27.51 |
| 2 | Second worker | 800.24 | 12.15 | 6.39 | 8.93 | 27.48 |
| Total value | | 1786.6 | 24.31 | 12.83 | 17.85 | 54.99 |
| Changed technological process (according to the second set of actions) | | | | | | |
| 1 | First worker | 960.08 | 12.91 | 12.14 | 10.10 | 35.15 |
| 2 | Second worker | 731.44 | 12.90 | 12.10 | 10.11 | 35.11 |
| Total value | | 1691.52 | 25.81 | 24.24 | 20.21 | 70.26 |
| Changed technological process (according to the third set of actions) | | | | | | |
| 1 | First worker | 1027.64 | 12.91 | 12.32 | 9.53 | 34.76 |
| 2 | Second worker | 887 | 12.90 | 12.29 | 9.54 | 34.73 |
| Total value | | 1914.64 | 25.81 | 24.61 | 19.07 | 69.49 |

To assess the effect of the implemented actions in relation to the changed technological process, elements of qualitative and quantitative description were applied (the second stage of the procedure), and the calculation of the key characteristics of the process was performed.

Figure 8 shows the results of the value stream mapping for the changed technological process. By applying a set of measures, the interaction scheme at the production site is optimized. The technological operation for the overhanding of 3–5 mm FPC was eliminated. During the formation of the finish layer, the operation for alignment of joints of metal structures is excluded due to the lack of value of this measure. The operation of alignment of the fire protection layer for aesthetic appearance is excluded due to the lack of value of this measure. Work planning was performed in accordance with the test schedule to exclude organizational and technological interruptions and displacement of workers to other areas. Compilation of competence matrix, introduction of mentoring, and delegation of powers to foremen have increased the efficiency of FPC work. To eliminate unplanned stoppages in the process, the purchase of additional apparatuses, such as a sandblaster, has been made. The main operations associated with the fire protection application process were taken into account when mapping the value creation flow. Thus, the efficiency of the process has increased by 29%.

Figure 9 shows the results of the "Spaghetti" diagram for the individual elements of the changed technological process. As a result of the implementation of the concept of lean construction, unnecessary movements for communications of workers in the process were eliminated and places of storage of materials were identified. Additionally, the movements were minimized, which allowed us to reduce the time of the technological process.

The calculations for the key process characteristics of the technological process are presented in Tables 6 and 7.

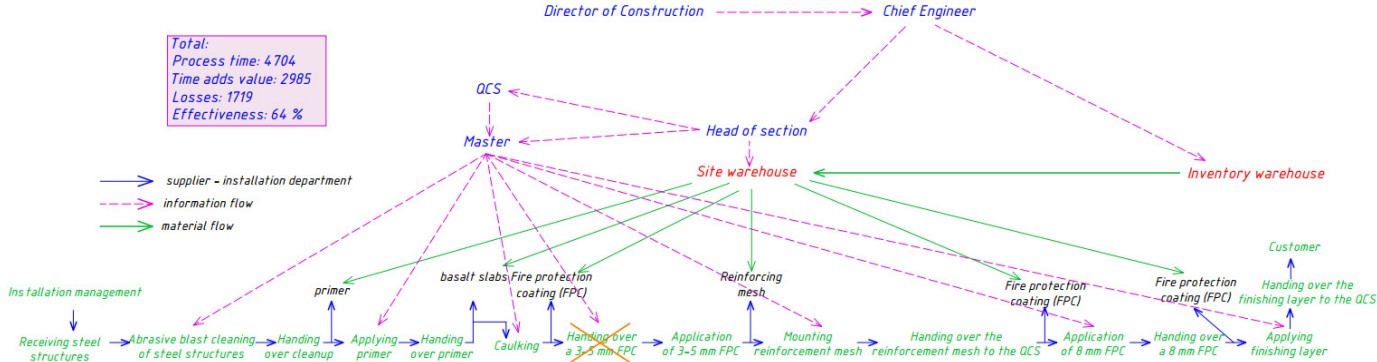

**Figure 8.** Results of value stream mapping for technological process changed according to the most preferred set of actions.

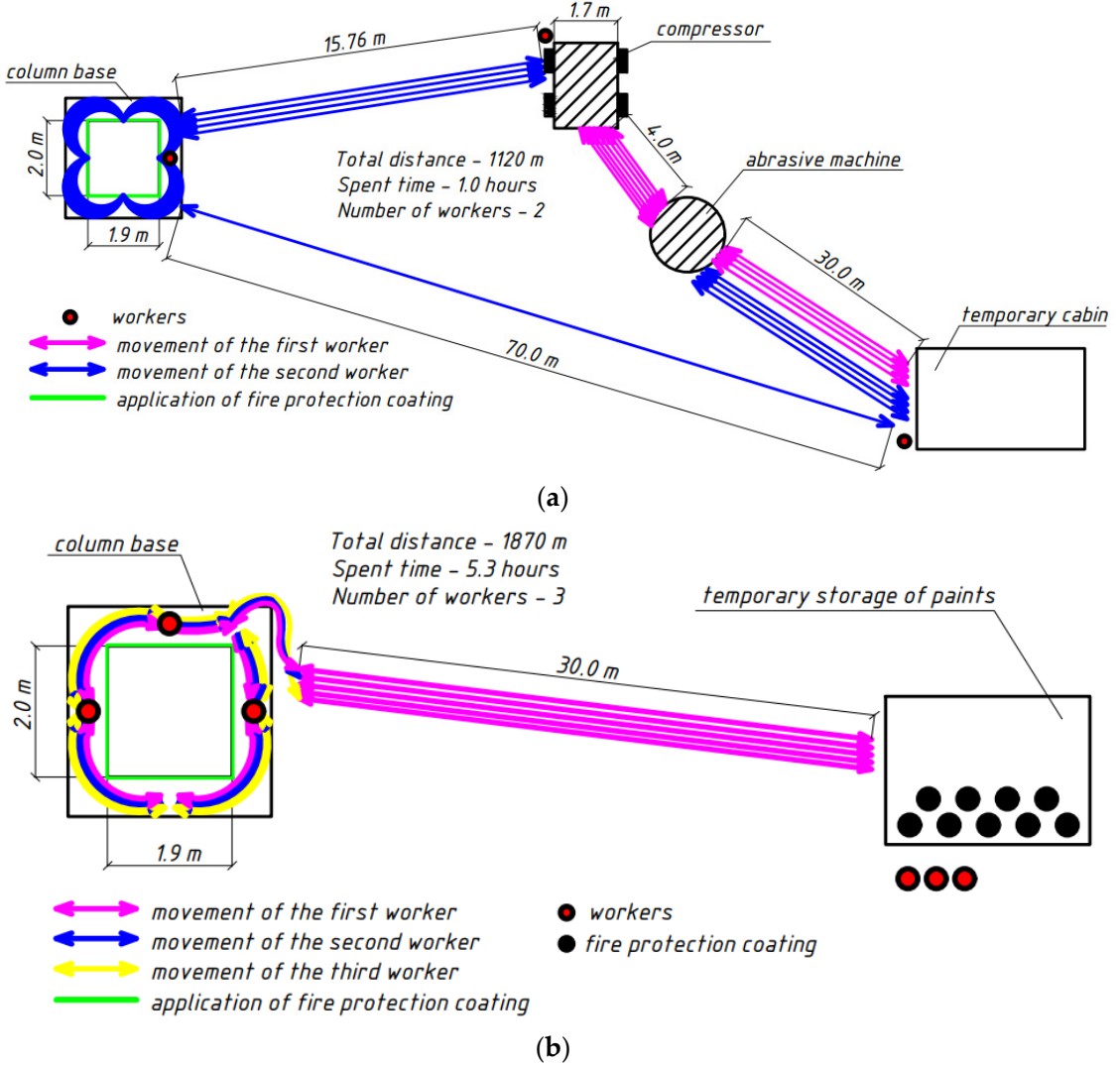

**Figure 9.** Results of drawing the "Spaghetti" diagrams for individual elements of the technological process, changed according to the most preferred set of actions: (**a**) procedure of abrasive cleaning and (**b**) procedure of supplying places of work with fire-protective epoxy coating from the area of temporary storage for coating materials.

**Table 6.** Timekeeping elements of the technological process, changed according to the most preferred set of actions.

| Categories of Process Operations | The Duration of Operations within the Technological Process with an Ordinal Number (See Table 2) | | | | | | | | | | | | | Total Duration of Operations in the Process |
|---|---|---|---|---|---|---|---|---|---|---|---|---|---|---|
| | 1 | 2 | 3 | 4 | 5 | 6 | 7 | 8 | 9 | 10 | 11 | 12 | 13 | |
| All operations | 501 | 160 | 110 | 150 | 271 | 973 | 0 | 96 | 60 | 2053 | 30 | 220 | 80 | **4704** |
| Operations that add value | 276 | 20 | 80 | 25 | 67 | 449 | 0 | 66 | 30 | 1712 | 20 | 220 | 20 | **2985** |
| Operations that do not add value | 225 | 140 | 30 | 125 | 204 | 524 | 0 | 30 | 30 | 341 | 10 | 0 | 60 | **1719** |

**Table 7.** The order of work in the application of FPC for the technological process changed according to the most preferred set of actions.

| No. | Procedure of the Technological Process | Number of Workers | Technological Equipment and Tools | Duration of Operations (the Result of Timekeeping) on 18 m$^2$, min | | |
|---|---|---|---|---|---|---|
| | | | | Operations That Add Value | Operations That Add Value | Operations That Add Value |
| 1 | Abrasive cleaning | 2 | Sandblaster | 276 | 225 | 501 |
| 2 | Handing over the cleaning to the QCS | 2 | | 20 | 140 | 160 |
| 3 | Priming | 1 | | 80 | 30 | 110 |
| 4 | Handing over the primer to the QCS | 1 | | 25 | 125 | 150 |
| 5 | Caulking of hard-to-reach places | 1 | Measuring tape, hacksaw | 67 | 204 | 271 |
| 6 | Application of 3–5 mm layer of FPC | 1 | Airless spraying unit "Wagner 950E" | 449 | 524 | 973 |
| 7 | Installation of reinforcing mesh | 3 | Measuring tape | 66 | 30 | 96 |
| 8 | Handing over the reinforcing mesh to the QCS | 1 | | 30 | 30 | 60 |
| 9 | Application of an 8 mm layer of FPC | 2 | Airless spraying unit "Wagner 950E" | 1712 | 341 | 2053 |
| 10 | Handing over the FPC to the QCS | 1 | | 20 | 10 | 30 |
| 11 | Application of the finish layer | 2 | | 220 | | 220 |
| 12 | Handing over the finish layer | 1 | | 20 | 60 | 80 |
| | Summary: | 17 | | **2985** | **1719** | **4704** |

　　　　Additionally in relation to the initial and changed (according to the most preferred set of actions) technological processes, the calculations of the following additional aggregate characteristics are made:

—　The average rate of production of one unit of workers (two workers) for a work shift of 10 h;
—　The number of workers required to implement the technological process in the context of individual calendar periods.

The calculated values are shown in Table 8 and Figure 10.

**Table 8.** Calculations of characteristics for the initial and changed (according to the most preferred set of actions) technological processes.

| No. | Name of Parameter | | Unit of Measure. | Value for the Technological Process | | Changing the Value | |
|---|---|---|---|---|---|---|---|
| | | | | Initial | Changed | Absolute | Relative |
| 1.1 | Duration of individual procedures within the technological process | Abrasive cleaning | min. | 785 | 501 | 284 | 36.18 |
| 1.2 | | Priming | min. | 450 | 110 | 340 | 75.56 |
| 1.3 | | Caulking | min. | 422 | 271 | 151 | 35.78 |
| 1.4 | | Application of 1–2 layers of FPC | min. | 1409 | 973 | 436 | 30.94 |
| 1.5 | | Mesh installation | min. | 120 | 96 | 24 | 20.00 |
| 1.6 | | Application of a layer of 8 mm FPC | min. | 2139 | 2053 | 86 | 4.02 |
| 1.7 | | Final painting | min. | 240 | 220 | 20 | 8.33 |
| 2.1 | Total duration of the technological process operations | all operations | min. | 7265 | 4704 | 2561 | 35.25 |
| 2.2 | | Operations that add value | min. | 3045 | 2985 | 60 | 1.97 |
| 2.3 | | Operations that do not add value | min. | 4220 | 1719 | 2501 | 59.27 |
| 3.1 | The average rate of production of one unit per work shift (10 h) | Application of fire protection to ensure the fire resistance limit of R120 | $m^2$/(units. days) | 2.27 | 3.00 | 0.73 | 32.16 |
| 3.2 | | Restoration of corrosion protection) | $m^2$/(units. days) | 13.45 | 17.90 | 4.45 | 33.09 |
| 3.3 | | Finishing coating of corrosion protection | $m^2$/(units. days) | 12.00 | 12.00 | 0 | 0.00 |

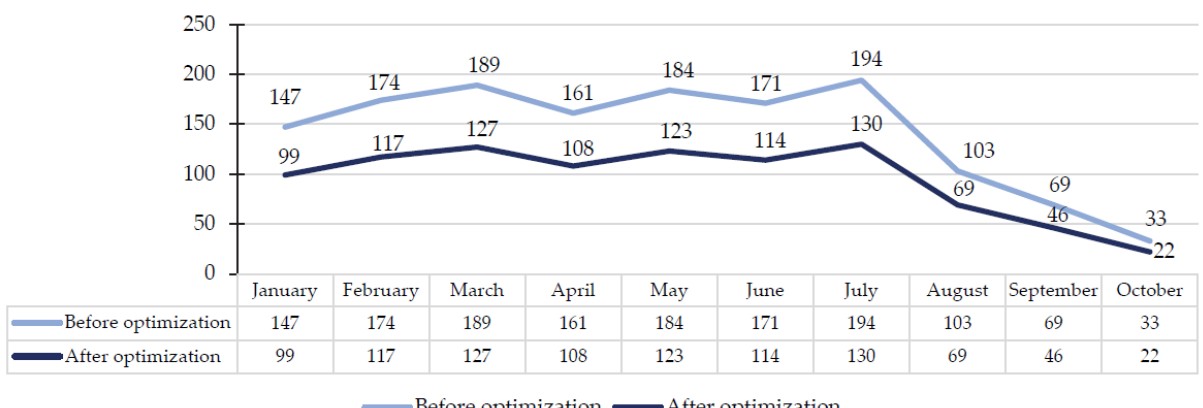

**Figure 10.** Calculations of the required numbers of workers for individual calendar periods.

Thus, the effect of applying the elements of the concept of lean construction in relation to the technological process of applying FPC is determined by the following points:

— Increase in output per employee for application of FPC and restoration of corrosion protection by approximately 33%;
— Reduction of the total duration of the technological process by approximately 35%.

Thus, the availability of the results of the procedure and the compliance of the results with the conditions of the implementation processes of the elements of the concept of lean construction in the activities of oil and gas organizations demonstrates the high practical significance of the developed procedure.

The developed procedure has distinctive features that limit its field of effective application:

1. In the implementation of measures to reduce the duration of the technological process, the impacts on the total cost and the quality of the performed work were not

considered. To account for all three parameters of the technological process, the method of linear convolution can be used.

2. There is lack of consideration of the indicator of change in the total duration of the technological process in substantiating the effect of the implementation of each alternative set of measures to improve the technological process. This disadvantage of the developed methodology can be eliminated by describing the interaction between indicators of the duration of individual operations of the technological process and the indicator of the total duration of the technological process. The development of an appropriate mathematical model is planned for the future stages of the study.

## 4. Conclusions

The following results were obtained in the study:

1. The review and analysis of scientific works were conducted in the field of implementing the concept of lean construction in the activities of industrial enterprises, including companies of the oil and gas complex. It is concluded that there are no scientific developments aimed at implementing the concept of lean construction to improve the technological process for applying FPC on the surface of metal structures as part of the oil and gas facility.

2. A procedure for applying the concept of lean construction was developed and implemented as a practical example to improve the technological process of applying an FPC on the surface of metal structures as part of the production facility of the oil and gas complex. A distinctive feature of the procedure is the formation of alternative sets of actions and the subsequent selection of the preferred set of actions by the criterion of the ratio of the effect from the implementation of activities to the corresponding required amount of cash resources.

The theoretical significance of the results of the study consists of the development of methodological approaches and improvement of tools in the field of implementation of the concept of lean construction in the activities of specialized organizations of the oil and gas complex, implementing critical technological processes that determine the operational reliability of construction projects.

The practical significance of the results of the study consists of the possibility of using the relevant scientific developments in the activities of construction organizations to improve technological processes implemented not only in the stage of construction framework construction (the study considered the process of applying FPC on the surface of building structures) but also during the stages of excavation, foundation device, laying utilities, etc.

Based on the results of the study, the requirements are formulated, which should satisfy the technological process, considered as an object of implementation of the lean construction concept, to ensure the effective implementation of the developed procedure. The mentioned requirements include the following:

1. A high degree of influence of the efficiency of the technological process implementation on the quality indicators of construction products, including those from the perspective of ensuring the operational reliability of construction projects;

2. The inseparability of the technological process in relation to the content of construction projects as well as the identity or high repeatability of the structure of the technological process in the context of various construction projects;

3. The flow or rhythmic nature of the technological process in the construction of a separate object (group of objects) as well as the involvement of significant amounts of resources (labor, machinery, and other factors) in the implementation of the technological process in the construction of a separate object (group of objects);

4. Availability of a limited staff of specialists involved in the implementation of the technological process during the construction of a separate object or a series of similar (in terms of a certain set of features) objects;

5. A certain level of complexity of the structure of the technological process under consideration, determining the presence of different categories of losses, and minimizing or eliminating them through the development and implementation of various measures;

6. The possibility of adequate assessment of the impact of the results of the implementation of measures to improve the technological process on the spatial and temporal characteristics of the technological process, and the possibility of adequate assessment of the timing of the implementation of measures as well as the required amount of resources;

7. The presence of known limited amounts of resources allocated for the implementation of measures to transform the technological process under consideration and, as a consequence, the need to prioritize activities and form the most preferable way of their implementation within a limited time interval.

**Author Contributions:** Conceptualization, M.G.; methodology, A.R. and D.S.; formal analysis, A.R.; investigation, N.I.; resources, N.I.; data curation, M.G. and D.S. All authors have read and agreed to the published version of the manuscript.

**Funding:** This research was funded by the Ministry of Science and Higher Education of the Russian Federation within the framework of the state assignment No. 075-03-2022-010 on 14 January 2022 (Additional agreement 075-03-2022-010/10 from 9 November 2022).

**Data Availability Statement:** Not applicable.

**Conflicts of Interest:** The authors declare no conflict of interest.

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
