# Peer review of "Implementation of Elements of the Concept of Lean Construction in the Fire Protection of Steel Structures at Oil and Gas Facilities"

_buildings, doi:10.3390/buildings12112016_

Round 1

Reviewer 1 Report

The paper studies the implementation of elements of the concept of lean construction in the fire protection of steel structures at oil and gas facilities.The rest of the paper is well structured and illustrated. There are, however, several points regarding the content of the paper that should be either improved upon or clarified:

(1) In the abstract, the research significance needs to be given after the research results.

(2) The research problem of the paper is not clear.

(3)The ideas in the discussion section of the paper are not very clear, even without a concise theme, and there is no comparison with other scholars' views.

(4)The stated contribution is not clear. The author should highlight the contribution in the     revision. The Introduction section should list the research objectives and research scope. In addition, the limitations were not well discussed.

(5) Figures 6-9 should be further explained in more details.

Reviewer 2 Report

The authors present a very interesting and well-structured technically-strong research. Congratulations on that. Nevertheless, some points that could increase the overall quality of the research.

1-     In the abstract is missing the main objective of this research – which could be defined by the authors to close a research gap regarding the concept of lean construction in the fire protection of steel structures at oil and gas facilities (based on the literature review from the authors in lines 83-85). Still, the English writing-style should be improved to create a sense of interest by the potential reader.

2-     In the conclusions (and in the abstract) only the benefits of the application of the concept of lean construction in the fire protection of steel structures at oil and gas facilities. However, the limitations and difficulties that the authors encountered across the research should also be addressed. These may be in terms of the research itself or based on literature review.

3-     The Images of Figure 2 should be enlarged. These are very important Figures in the presented research and should have a better (higher) definition (resolution). Still, these should be named as a) and b).

4-     In Figure 3 to keep the consistency, the step number 5 is missing. The authors jump from the 4 to the 5.1 and so on. When analyzing the diagram, the reader misses what the step 5 stands from. Please reorganize.

5-     The images in Figure 4 should also be rearranged based on the suggestions over the images of Figure 3 in the previous point.

6-     In the conclusions section is missing the clear distinction between academic and managerial implications of the presented research. I suggest the authors to create a subchapter under chapter 4 where the managerial and academics implications are clearly illustrated.

7-     Finally, the suggestions for further research are also missing. For example, the authors could argue that despite there is a huge amount of research on the successful implementation of the concept of lean construction in the activities of construction organizations (as the authors document in lines 83-89), it could be argued that your research can cover the identified gap, namely

Good job

Regards

Round 2

Reviewer 2 Report

I am satisfied with the changes done by the authors.

Well done!
